# A Technique for Centrifugal Pump Fault Detection and Identification Based on a Novel Fault-Specific Mann–Whitney Test

**DOI:** 10.3390/s23229090

**Published:** 2023-11-10

**Authors:** Zahoor Ahmad, Jae-Young Kim, Jong-Myon Kim

**Affiliations:** 1Department of Electrical, Electronic and Computer Engineering, University of Ulsan, Ulsan 44610, Republic of Korea; zahooruou@mail.ulsan.ac.kr; 2Prognosis and Diagnostics Technologies Co., Ulsan 44610, Republic of Korea

**Keywords:** vibration signals, soft faults, fault detection and identification, centrifugal pump

## Abstract

This work presents a technique for fault detection and identification in centrifugal pumps (CPs) using a novel fault-specific Mann–Whitney test (FSU Test) and K-nearest neighbor (KNN) classification algorithm. Traditional fault indicators, such as the mean, peak, root mean square, and impulse factor, lack sensitivity in detecting incipient faults. Furthermore, for defect identification, supervised models rely on pre-existing knowledge about pump defects for training purposes. To address these concerns, a new centrifugal pump fault indicator (CPFI) that does not rely on previous knowledge is developed based on a novel fault-specific Mann–Whitney test. The new fault indicator is obtained by decomposing the vibration signature (VS) of the centrifugal pump hierarchically into its respective time-frequency representation using the wavelet packet transform (WPT) in the first step. The node containing the fault-specific frequency band is selected, and the Mann–Whitney test statistic is calculated from it. The combination of hierarchical decomposition of the vibration signal for fault-specific frequency band selection and the Mann–Whitney test form the new fault-specific Mann–Whitney test. The test output statistic yields the centrifugal pump fault indicator, which shows sensitivity toward the health condition of the centrifugal pump. This indicator changes according to the working conditions of the centrifugal pump. To further enhance fault detection, a new effect ratio (ER) is introduced. The KNN algorithm is employed to classify the fault type, resulting in promising improvements in fault classification accuracy, particularly under variable operating conditions.

## 1. Introduction

Worldwide growth in the population has increased the demand for agricultural production and water supply management, which provides the primary means for human survival, food security, nutrition, and economic growth. CPs can play an important role in drainage, water supply, and agricultural productivity [1]. CP refers to a pump that transforms electrical energy into mechanical energy, utilizing impellers to facilitate the transportation of fluids [2]. CPs are commonly used for irrigation, supplying water to livestock, drainage, and industrial fluid control [3,4]. However, CPs operate in harsh conditions for extended durations, including intense temperatures, gritty environments, and variable operating speeds, which makes them vulnerable to defects. Defective CPs can lead to economic losses, waste of resources, catastrophic failures, environmental pollution, and threats to human safety [5]. Therefore, the timely detection and identification of faults in the CP are of primary significance. Faults in the CP can be classified into hard and soft faults (SFs). Hard defects can be identified by humans; in this condition, the pump stops its operation at once. The main reasons for these defects include a broken shaft, bearing, or impeller. In contrast, SFs in the CP progressively affect the operating efficiency. These defects can be very dangerous, as they may not be noticed by humans [6]. Therefore, the development of a technique based on artificial intelligence (AI) for early detection and identification of SFs could be very advantageous. Defects related to mechanical seals account for 34% of SFs in these pumps and have severe consequences in the form of fretting, shaft denting, and wear [7]. Furthermore, the complex flow of fluid in the pump can cause SFs in the impeller, which can lead to mechanical and fluid flow-related defects in the CPs. In this study, to avoid unexpected downtime and increase the healthy operation life of the CPs, SFs related to a mechanical seal (e.g., mechanical seal holes (MS-Hs), mechanical seal scratches (MS-Ss), and impeller defects (IDs)) are considered.

Non-destructive testing methods, including orbit analysis [8], deflection shape analysis [9], acoustic emission (AE) analysis [10], and vibration signal (VS)-based monitoring, are used for the intelligent fault diagnosis of CPs [11]. A fault in the mechanical specimen will change the stiffness of the material, which will result in a shock in the VS that appears at a specific frequency, providing an overall picture of the status of the machine. Furthermore, VS-based analysis permits early fault detection in real time with high sensitivity [12]. These unique advantages of VS-based monitoring make it attractive for the industry compared to other non-destructive testing methods. Hence, in this work, VS is utilized for the detection and identification of SFs in CPs.

### Related Research Studies

A fault in a CP results in abnormal vibration, which is detected by an attached accelerometer. The VS acquired from the CP is complex and non-stationary; therefore, extracting health-sensitive information from these signals can help AI algorithms identify if the health state of CPs requires preprocessing in the temporal domain (TD), time-frequency domain (TFD), and Frequency domain (FD) [13,14]. Wang et al. [15] introduced a dual-phase approach for SF classification in pump rotating parts. First, to process the VS, the dot products of each time-series pair of VS points are used to create a Gramian matrix. Following this, the matrix is normalized between zero and one. The normalized matrix is used to calculate the Gramian angular field by applying the inverse cosine function to each element of the normalized matrix. These fields are then provided to recurrent neural networks (NNs) having dual-stage attention mechanisms for the identification of SFs in the second stage. The VS acquired from the CPs contains background interference noise due to their operation in harsh working conditions. Therefore, to overcome the noise in the VS, the time domain generalized autocorrelation function can be used [16]. A fault in the CP will result in an abnormal shock, which will appear at a particular frequency in the FD. Therefore, analyzing the TD and FD of the VS can be used for the detection and identification of SFs in CPs with the help of AI techniques, such as approaches using artificial NN, recurrent NN, a support vector machine (SVM), and KNN algorithms [17,18,19]. Ferracuti et al. [20] proposed a hybrid technique that combines TD and FD analysis for the fault diagnosis of rotating machines. The proposed work represents the VS on multiple time scales. From each scale, a rotating machine health-sensitive indicator is calculated using the Wasserstein distance. To harness fault-related non-stationary variations in the VS for CP health diagnosis, TFD techniques such as WPT, variational mode decomposition (VMD), and empirical mode decomposition (EMD) can be employed [21,22,23]. Tiwari et al. [24] identified the health conditions of a CP using WPT and SVM. Ahmad et al. [25] extracted statistical indicators from the VS of the CP in the TD, FD, and TFD. The extracted indicators were merged into one feature vector, and the operating conditions of the CP were identified using SVM. Statistical features obtained from the VS of the CP usually resulted in a higher dimension feature pool. Furthermore, it is better to use fewer features to represent the health state of the CP. For this reason, feature selection and dimensionality reduction techniques are introduced [26,27]. Dong et al. [28] utilized improved particle swarm optimization for selecting the intrinsic mode function of VMD. The selected intrinsic mode function is used for feature extraction. Researchers have also been focusing on deriving a fault indicator from the VS of pumps, bearings, gears, and pipelines that have a direct relationship with the health of the specimen. Li et al. [29] extracted fault-related indicators from 1D VS using expansion operation and deep NN. Jiao et al. [30] extracted a multi-scale entropy-based energy moment indicator as a fault indicator from the VS of the bearing under normal and defective conditions. The least-square SVM identified the working condition of the bearing by processing the proposed fault indicator. Rai et al. [31] calculated the pipeline health state indicator independent of prior knowledge using the multi-scale Kolmogorov–Smirnov (KS) test. Similarly, pipeline leak fault indicators are developed using the Mann–Whitney (U-test) test on multiple time scales [32].

The aforementioned techniques can identify SFs in CPs. However, there exist several limitations. First, the statistical features, such as the mean, variance, root mean square, etc., obtained from the VS in the TD are not susceptible to SFs. Furthermore, to select SF-sensitive features, extensive feature preprocessing is needed [33,34]. Second, the FD considers signal statistical properties invariant over time; however, the VS of the CP in defective operating conditions is non-stationary, which makes FD analysis less attractive [35]. TFD techniques, such as EMD and wavelet transform techniques, can be used to obtain time-varying fault-related transients in the signals for training supervised AI techniques for fault classification in CPs. Supervised techniques need a huge amount of data for effective training. However, obtaining a significant amount of training data from defective CPs is not feasible due to potential risks to human safety and a waste of resources. Moreover, training supervised AI algorithms with an insufficient amount of training data can result in false alarms regarding CP health. The multi-scale U-test and KS test can detect faults in CPs independent of prior knowledge. The multi-scale analysis obtains coarse-grained information about the signal over multiple time scales [36]. However, this represents the signal on small scales, reducing its length and leading to imprecise analysis. Furthermore, the selection of time scales requires thorough experimentation and domain expertise [37]. To address the above-mentioned concerns, a CPFI is developed based on a novel FSU test in this study. The new fault indicator is obtained by decomposing the VS of the CP hierarchically into its respective time-frequency representation using the WPT. The nodes of the WPT represent specific frequency bands at different time scales. Therefore, the node containing the fault-specific frequency band is selected, and the U-test statistic is calculated from it. Combining the hierarchical decomposition of the VS for fault-specific frequency band selection and the U-test yields the new FSU test. The test output statistic results in the CPFI, which shows sensitivity toward the health condition of the CP and varies based on the operating conditions of the pump. To assess the changes in the fault indicator due to the working conditions of the CP, a new ER is developed for fault detection in the pump, irrespective of previous knowledge. Finally, to identify SFs in the CP, the fault indicator is classified using the KNN algorithm.

The novelty and uniqueness of this study can be outlined as follows. (i) A new CP health-sensitive indicator, i.e., CPFI, is developed using the novel FSU test. (ii) To detect SFs in the CP, ER is introduced that assesses changes in the CPFI due to the operating conditions of the pump, irrespective of previous knowledge. Additionally, the significant aspects of this study are as follows. (i) The FSU output statistic is used as a CPFI. (ii) SF detection and identification in the CP is performed using a novel FSU test and KNN classification algorithm. (iii) The effectiveness of the proposed techniques is confirmed through the utilization of real-world CP vibration data.

This work is structured as follows: Section 2 elucidates the fundamental concepts employed in this study. The steps in the proposed technique are presented in Section 3. The results obtained from the proposed methodology are presented in Section 4. Lastly, Section 5 encapsulates the conclusion of this study.

## 2. Fundamental Concepts

### 2.1. Mann–Whitney U-Test

Let A(x) and B(x) be the two independent samples drawn from a 1D distribution. Here, the U-test assesses whether the two samples originate from the same or distinct populations regardless of their distribution. The U-test hypothesis for the assessment of the samples can be conceived in the following manner:

(i) Zero hypothesis (H0): The H0 will be valid if there is no significant difference in the U-test statistics of the two samples, A(x)=B(x);

(ii) Alternative hypothesis (H1): The H1 will be valid if the U-test statistics of A(x) increase or decrease in comparison to B(x), A(x) ≠ B(x).

The proposed CPFI can be obtained using Equation (1) from the U-test statistics.
(1)$$CPFI_{i}=\frac{FSU_{i}-\mu_{i}}{\sigma_{i}}$$

The sample mean and variance are represented by $\mu_{i}$ and $\delta_{i}$ in Equation (1). The ${FSU}_{i}$ can be obtained from Equation (2).
(2)$$FSU_{i}=s_{1}s_{2}+\frac{s_{i}(s_{i}+1)}{2}-\sum rank_{i},$$

In Equation (2), $s_{1}$ and $s_{2}$ represent the two independent 1D samples, and the sum of rank for the sample of interest is represented by ${rank}_{i}$. A change in the health condition of the CP will affect the CPFI given in Equation (1), which can be used to identify the working state of the CP. From Equation (1), it can be concluded that, for a CP under normal conditions, there will be no significant change in the CPFI. Thus, H0 will be valid. In contrast, H1 will hold true if the CP operating state shifts from a healthy to a defective operational state. In this study, to harness the changes in the CPFI due to the operational conditions of the CP for the detection of SF without previous knowledge, a new ratio called the ER is used.

### 2.2. Wavelet Packet Transform

The time-frequency representation of a 1D time domain signal can be obtained using a WPT. The WPT hierarchically decomposes a 1D signal into *l* levels using a combination of high-pass and low-pass filters, creating *2l* nodes at each *l*. Every level of the WPT corresponds to a frequency range that is double the width of the preceding level and half the width of the subsequent one. The mathematical representation of the WPT coefficients is provided in Equations (3) and (4).
(3)$$C_{j+1}^{2n}[m]=\sqrt{2}\sum_{}^{}{}_{l}P_{l-2m}C_{j}^{n}[m]$$
(4)$$C_{j+1}^{2n+1}[m]=\sqrt{2}\sum_{}^{}{}_{l}L_{l-2m}C_{j}^{n}[m]$$

The detailed and approximate coefficients obtained from the high-pass $P_{l-2m}$ and low-pass $L_{l-2m}$ filters are represented by $C_{j+m}^{2n}$ and $C_{j+m}^{2n+1}$ in Equations (3) and (4), respectively.

The multi-scale analysis of a 1D VS provides information about the low-frequency components of the signal. However, a comprehensive multiresolution analysis of the 1D VS can be obtained from WPT in both lower and higher frequencies. This overcomes the limitations of multi-scale analysis. For this reason, in this work, the WPT is employed to derive a complete time-frequency depiction of the input VS. Furthermore, the fault-specific frequencies of the CP are calculated, and the WPT nodes (according to these frequencies) are selected for extraction of the new CPFI.

## 3. Proposed Technique

The technique proposed in this study starts with the acquisition of the VS from the CP and ends with the classification of SFs in the CP. Figure 1. provides a graphical illustration of the steps involved in the proposed technique. These steps can be explained as follows.

Step 1: VSs are acquired from the CP during both normal and defective operating conditions. The defective operating conditions considered in this study are MS-H, MS-S, and ID.

Step 2: An SF in the CP will produce irregular impulses in the VS, which will appear at specific frequencies in the spectrum. These frequencies can help extract CP health-sensitive information from the signal. In this step, the VSs are decomposed into independent frequency bands using the WPT up to “*l*” levels with the help of Equations (3) and (4) explained in Section 2.2. The WPT nodes at level “*l*” (where the fault-specific frequencies can be perceived) are selected, and the U-test explained in Section 2.1 is applied. Combining the WPT node selection process and the U-test yields a new FSU test. The FSU output statistics result in the new CPFI, which can be calculated using Equation (1).

Step 3: Whenever the operating conditions of the CP switch from a healthy to a faulty state, the defective frequencies occurring in the spectrum will change the distribution of the frequency spectrum. Since the FSU test, unlike the MWU test, considers only fault-specific frequencies, there will be a significant change in the CPFI. To utilize the changes in CPFI caused by the working conditions of the CP for SF detection independent of previous knowledge, the new *ER* is calculated using Equation (5).
(5)$$ER=\frac{CPFI}{\sqrt{s}}$$

In Equation (5), the *CPFI* is the fault indicator obtained from Equation (1), and *s* is the total number of samples.

Step 4: To identify the centrifugal pump’s operational conditions, the CPFI acquired in step 3 is utilized as input for the classification through the KNN algorithm. The KNN classification algorithm is selected in this work due to its uncomplicated structure and minimal computational cost for classification tasks.

## 4. Results and Discussion

The results obtained from the proposed technique for SF detection and classification in CPs are discussed in this section. To ensure an unbiased evaluation of the proposed approach, the results will be explained in two distinct subsections. The first section will provide a comparison of the proposed CPFI obtained from the FSU test with traditional features, and the second section will offer a comprehensive comparison of the classification capabilities of the proposed method with those of the reference state-of-the-art methods.

### 4.1. Data Acquisition Setup

Real-world CP VSs are used for the validation of the proposed technique. The schematics and pictorial view of the test setup for VS acquisition are given in Figure 2a,b. The test setup consists of three main parts: (1) a multistage CP (PMT-4008) having five impellers fitted in series and powered by a 5.5-kW motor; (2) a pipeline system with pressure gauges, a main tank, and a secondary tank; and (3) a display panel that controls the power switch, speed, flow rate, water supply, and temperature. The main purpose of the CP was to lift water from one tank (main) to another secondary tank, which was placed at a height of 15 m.

#### 4.1.1. System Development for Data Acquisition

As displayed in Figure 2c, the acquisition process was initialized by recording vibration data at a constant 1733 rpm using a 622b01 accelerometer. Afterward, the raw signal from this sensor was processed with an analog-to-digital converter (ADC), which was integrated into a National Instrument (NI) 9234 module and transmitted through a universal standard bus to a personal computer where the digital vibration data were stored.

The shaft in the CP serves as a wave propagation path in the pump. Therefore, the accelerometer was affixed to the pump using adhesive positioned close to the impellers along the axial direction. After attaching the accelerometer, the data acquisition process was controlled with self-developed software using libraries from NI and Python. Prior to data acquisition, the accelerometer was calibrated, and its sensitivity was tested through hammering. After ensuring that the acquisition system functioned properly, a dataset was recorded under healthy and defective conditions from the CP.

#### 4.1.2. Data Acquisition and Dataset Description

CPs operate in harsh conditions for extended durations, including intense temperatures, dusty environments, and variable operating speeds, which makes them vulnerable to SFs. A mechanical seal (MS) consists of a rotating and stationary part connected by a spring. The spring applies pressure on the face of the MS so that the two components remain in contact. However, a dirt particle can get trapped between the seal faces, leading to SFs-like holes and scratches on the sealing faces. For this reason, in this study, SFs due to MS-Hs and MS-Ss are considered. To elucidate the MS-H defective condition, a hole of 28 × 28 mm (diameter × depth) was created with a drilling machine on the rotating part of the MS. For MS-S defects, a scratch with a 38 mm inner diameter was made with a drilling machine in the rotating part of the MS.

The complex fluid flow inside the pump can cause corrosion in the impeller. Crevice corrosion is a type of corrosion in which multiple connected holes appear on the impeller surface. This is the most common defect type that affects the health of the impeller. The same kind of SF is replicated in this study to represent an ID; out of the five cast iron impellers (161 mm diameter), a metallic portion was removed from the fifth, leaving a fault size of 2.5 × 2.8 mm (diameter × depth) to replicate the influence of crevice corrosion.

After creating defects in the impeller and MS, as shown in Figure 2d, VSs were obtained at a sampling rate of 25.6 kHz over 300 s for the normal and defective health states of the CP. The health states of the CP investigated in this study are normal and defective (i.e., MS-H, MS-S, and ID). Two datasets were recorded from the CP. The first dataset (D1) consisted of data collected under each operating condition at a pressure level of 3 bar at the pump inlet. For the second dataset (D2), the pressure at the pump inlet was increased to 4 bar, and data were recorded from the CP at each operating condition. The measurement noise levels for the vibration signals related to MS-H, MS-S, and ID were determined to be 69.10 dB, 62.07 dB, and 63.78 dB, respectively. Table 1 offers a comprehensive description of the datasets, and Figure 3 illustrates the VSs collected from the CP during both normal and faulty operating conditions.

### 4.2. Comparison of the Proposed CPFI with Traditional Features

Generated frequencies (GFs), electronic frequencies, and excitation frequencies (EFs) are of interest for CP fault detection and identification. The GFs in the spectrum show an imbalance, which is an important feature for identifying IDs in the CP. The GFs can be calculated using Equation (6).
(6)ID=n×H,
where *n* represents the harmonics and H represents the rpm (Hz) of the CP. Following the ISO 13373-1:2002 standard [38] for vibration condition monitoring, the ID frequencies are calculated up to the fifth harmonic, as shown in Figure 4. The yellow shading in Figure 4a shows the vibration spectrum of the CP in a healthy state, while the red shading in Figure 4b shows the vibration spectrum of the CP working under ID. The amplitudes of the third to fifth harmonics in ID conditions were increased by a factor of two compared to the harmonics of the defect-free condition.

The EF is an important indicator in the frequency spectrum for determining the health state of MS. The EF can be defined as the single highest-amplitude impulse in the spectrum. A change in the health condition of the MS will change the EF of the CP. To identify the change in EF due to MS health, the natural frequency in the flexural mode of vibration for MS can be computed along with its corresponding modes of vibration (MOVs) as follows:(7)$${MS}_{f}=\left(\frac{j\left(j^{2}-1\right)}{\pi d^{2}\sqrt{j^{2}+1}}\right)\sqrt{\frac{Fr^{2}}{3\rho }},$$
where *d* represents the ring diameter, *j* stands for the MOVs (from 1 to 5), *ρ* is the material density, and *r* refers to the ring thickness. The excitation frequency was altered upon the occurrence of an MS defect, as demonstrated in Figure 5. This figure also shows a minor amplitude increase in the excitation-frequency-related impulse, which resides between the flexural vibration’s first two modes under MSH, as can be seen in Figure 5b. Regarding MSS, an increase can also be witnessed in the amplitude of the EF between the second and third flexural MOV, as depicted in Figure 5c. In Figure 5, the yellow-highlighted frequency spectrum shows the CP in a healthy state, and the red-highlighted frequency spectrum shows the defective CP. The blue dotted lines in Figure 5 represent the second and third MOVs.

The frequency spectrum analysis of the CP presented in Figure 4 and Figure 5 provides information about fault-specific frequency bands. From Figure 4 and Figure 5, the frequencies up to the third MOV are important for the diagnosis of SFs. Higher frequencies occurring in the fourth and fifth MOVs are due to microstructural vibration noise and can be filtered out prior to fault indicator extraction from the VS. For this reason, in this study, the VSs are decomposed into independent frequency bands using the WPT up to *l* = *2*. Murlidharan et al. [39] evaluated different families of wavelets for fault diagnosis of CP and concluded the db4 wavelet to be outstanding in comparison to the rest. Therefore, the db4 wavelet is adopted for the decomposition of the VS in this study. The first two WPT nodes at level “*l*” are selected because they contain the frequencies up to the third MOV. Then, the FSU test is applied, and the corresponding CPFI is obtained.

The new CPFI is compared with the most commonly used TD indicators, such as the mean, peak factor, RMS factor, and impulse factor. The CPFI and the reference TD indicators obtained from D-1 and D-2 are shown in Figure 6 and Figure 7. The yellow highlighted region in the figures shows the CP in a healthy state, while the purple highlighted region shows the defective operating state of the CP. The new CPFI obtained from the proposed FSU test changes according to the change in the working state of the CP. Furthermore, the CPFI shows sensitivity toward the SFs of the CP irrespective of the fluid pressure at the inlet. The sensitivity of the new CPFI toward the CP health state can be explained in the following manner. An SF in the CP will produce irregular impulses in the VS, which will appear at specific frequencies in the spectrum. These frequencies will change the distribution of the VS according to the health state of the CP. Based on this distribution change, the FSU test selects the fault-specific frequencies and then applies the Mann–Whitney U-test. Since the FSU test considers only the fault-specific frequencies, significant changes are noted in the CPFI according to the working conditions of the CP. Additionally, the ER was 0.02 when the CP operated in normal conditions but increased significantly to 0.08 in the defective state. This ratio provides help in detecting the health status of the CP irrespective of prior knowledge, an advantage compared to supervised AI methods. Furthermore, when the condition of the CP changes from normal to defective condition, the test yields *p*-values ranging from 0.02 to 0.03 for both the D-1 and D-2 datasets at samples 300 and 301 (*p* < 0.05). This rejection of the null hypothesis indicates that there is a significant change in the ongoing health condition of the CP. Evidence from Figure 6 and Figure 7 clearly shows that the CP condition shifts from normal to defective at sample number 300, a finding supported by the *p*-value of the test.

In comparison, the TD indicators do not show sensitivity toward the SFs in the CP; these vary almost the same as they do under normal operating conditions. These features show worse sensitivity when the pressure at the inlet is increased to 4 bar. The underperformance of these features is expected as they are not appropriate for SFs. Furthermore, the TD features are very sensitive to noise in the signal. Figure 6d shows that the RMS factor is sensitive to the working conditions of the CP; it increases slightly when the operating conditions of the CP change from a normal to a defective state. However, the RMS factor cannot differentiate between the different types of SFs occurring in the CP.

### 4.3. Validation of the Proposed Method for SF Classification in CPs

The capability of the proposed technique for classifying SFs in CPs is presented in comparison with current cutting-edge methods within this section. The reference methods selected for comparisons are WPT+PCA-MSVM and WPT+BE-MSVM, as presented in [24], and the method proposed by Cao et al. [12]. These methods are selected for comparison because they attempt to solve the same problem as that of the proposed method. Furthermore, the parameters adopted for the experimental setup presented in this study are similar to these aforementioned comparison methods. Both the proposed and reference methods use VS for fault diagnosis of CP. To make the evaluation fair, k-fold cross-validation (*k* = 3) is adopted with testing: training split of 80:20. The matrices used for comparing the performance of the proposed method against the reference methods are the true positive rate (TPR), overall accuracy (OA), average recall (AR), average precision (AP), and average error rate (AER). The mathematical equations for these matrices are given in Equations (8)–(11), where *A_TP_*, *A_FN_*, *A_FP_*, and *A_TN_* represent the true positives, false negatives, false positives, and true negatives, respectively. The TPR presented in Figure 8a is for D-1, while the TPR for D-2 is presented in Figure 8b. Table 2 shows the OA, AP, AR, and AER values for the reference techniques and proposed method. All the results displayed in Figure 8 and Table 2 are average values from 10 experimental trials.
(8)$$AR=\frac{1}{k}\sum_{i=1}^{k}\left(\frac{\left(A_{TP}^{i,m}\right)}{A_{TP}^{i,m}+A_{FN}^{i,m}}\right)\times 100(\%)$$
(9)$$AP_{}=\frac{1}{k}\sum_{i=1}^{k}\left(\frac{\left(A_{TP}^{i,m}\right)}{A_{TP}^{i,m}+A_{FP}^{i,m}}\right)\times 100(\%)$$
(10)$$AER=\frac{1}{k}\sum_{i=1}^{k}\left(\frac{\left(A_{TP}^{i,m})+(A_{FN}^{i,m}\right)}{(A_{TP}^{i,m})+(A_{FN}^{i,m})+A_{TN}^{i,m}+A_{FP}^{i,m}}\right)\times 100(\%)$$
(11)$$OA=\frac{1}{k}\sum_{i=1}^{k}\left(\sum_{m=1}^{L}\frac{A_{TP}^{i,m}}{A_{samples}}\right)\times 100(\%)$$

After applying the proposed technique to D-1, an overall TPR of 99.2% was obtained for the classification of SFs, with an AER of 1.56%, AP of 99.20%, and AR of 99.21%. For D-2, the proposed method resulted in an overall TPR of 99.24% with an AER of 1.43%. These results show that the proposed method can classify SFs in the CP irrespective of the fluid pressure at the pump inlet. The performance of the proposed method is due to its core idea of using the CPFI for training and classification purposes. Furthermore, the selection of health-sensitive frequencies for the FSU test and the lower sensitivity of the test to outliers also contributed to the improved performance of the proposed method. The new CPFI shown in Figure 6 and Figure 7 shows the ability to discriminate between the operating conditions of the CP. The classification accuracy of the classifier is directly related to the discriminance of the input data. Thus, the CPFI shown in Figure 6 and Figure 7 provides strong evidence that the proposed method can classify SFs accurately.

The reference method WPT+PCA-MSVM decomposes the VS into independent frequency bands using the WPT. After decomposing the signal, the WPT node is selected for TD statistical feature extraction using principal component analysis (PCA). The health-sensitive features are then selected with the wrapper method for identifying the health state of the CP. After applying the WPT+PCA-MSVM, an overall TPR of 96.3% with an AER of 7.5% for D-1 was obtained. For D-2, the reference method resulted in an overall TPR of 88.87% with an AER of 17.13%. For both datasets, the overall TPR was less than that of the proposed method because of the smaller information loss due to PCA. Furthermore, the TD features are less sensitive toward SFs, as in Figure 6 and Figure 7.

The WPT+BE-MSVM method continuously decomposes the VS into independent frequency bands using the WPT. After decomposing the signal, the WPT nodes are rearranged in order of energy, and significant TD features are extracted. From each node, features are extracted and classified using SVM. Based on the highest classification accuracy, the WPT node is suggested for fault diagnosis of a CP. With WPT+BE-MSVM, an overall TPR of 94% with an AER of 10.5% for D-1 was obtained. For D-2, the reference method resulted in an overall TPR of 92.04% with an AER of 13.29%. Once again, for both datasets, the overall TPR was less than that of the proposed method because the reference method selects WPT bases based on energy instead of fault-specific frequencies. Furthermore, TD features extracted from the VS are less sensitive toward SFs.

The method proposed by Cao et al. [12] first obtains a coarse-grain multi-scale representation of the VS and then extracts features using PCA. After extracting features, the method identifies the health state of the CP using the Gaussian mixture model. After applying the steps mentioned in [12], an overall TPR of 81.1% with an AER of 25.4.5% was obtained for D-1. For D-2, the reference method resulted in an overall TPR of 84.01% with an AER of 22.56%. For both datasets, the overall TPR was less than that of the proposed method because the reference method uses raw VS for feature extraction. As can be seen from Figure 6 and Figure 7, along with the fault characteristic frequencies, there was macrostructural vibration noise in the signal. This noise can affect the quality of the features extracted from the raw VS. This method underperformed due to the effects of heavy background noise.

The proposed CPFI is capable of detecting faults in the CP within 0.12 s when running on a PC equipped with an Intel^®^ Core™ i7-9700K CPU and 16 GB of RAM. With a processing time of under 1 s, the proposed method is well-suited for the detection of CP faults in real-time. The proposed method does not need any previous knowledge about the health state of the CP for SF detection. Furthermore, the implementation simplicity and low complexity for computation make the proposed method attractive for industrial applications to identify and classify SFs in CPs.

## 5. Conclusions

This work proposed a novel technique for fault detection and identification in centrifugal pumps based on the fault-specific Mann–Whitney Test and K-nearest neighbor classification algorithm. The vibration signals acquired from the centrifugal pump are hierarchically decomposed into their respective time-frequency bands using wavelet packet transform. Subsequently, the fault-specific frequency band is identified within this decomposition, and the Mann–Whitney test statistic is computed from it. The amalgamation of these processes forms the novel fault-specific Mann–Whitney test. The test output statistic serves as a centrifugal pump fault indicator. The fault indicator changes according to the change in the health condition of the centrifugal pump. To utilize this change for the detection of faults in the centrifugal pump effect ratio is used in this work. To identify defects in the centrifugal pump, the fault indicator is provided as an input to the K-nearest neighbor classification algorithm. The proposed method is validated on two different datasets obtained from a real-world centrifugal test bed at 3 and 4-bar inlet pressure. For both datasets, the proposed method resulted in an average fault classification accuracy of 99.20% and 99.24%, surpassing the performance of reference state-of-the-art methods. However, the centrifugal pump fault indicator, obtained using the proposed method, exhibits a high level of discrimination among different health conditions of the CP. Consequently, the presence of high variance could pose an issue when employing a classification algorithm other than K-NN for identifying the health conditions of the CP. In the future, this study will extend its application to diagnose fluid flow-related soft faults in centrifugal pumps, including issues such as cavitation and fluid leakage from the pump. Furthermore, the fault detection and classification performance of the proposed method under various types of noise will be verified in the future.

## Figures and Tables

**Figure 1 sensors-23-09090-f001:**
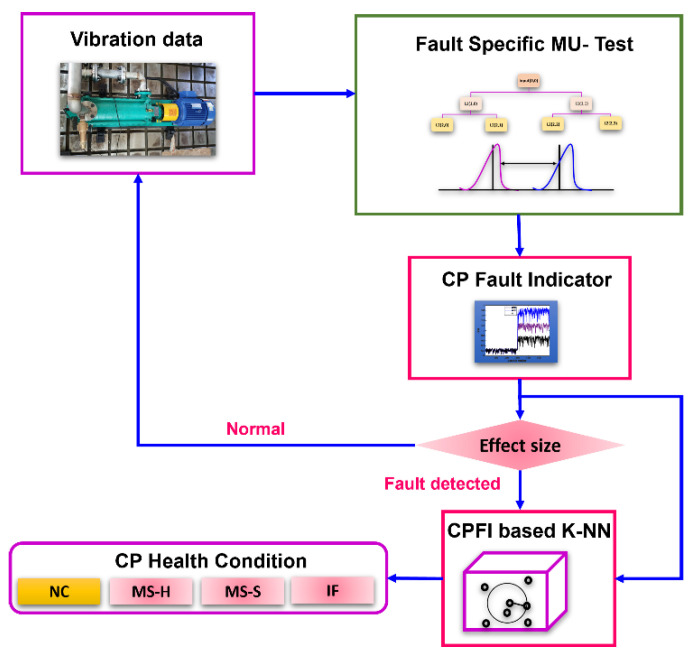
Graphical illustration of the proposed technique for detecting and classifying SFs in CPs.

**Figure 2 sensors-23-09090-f002:**
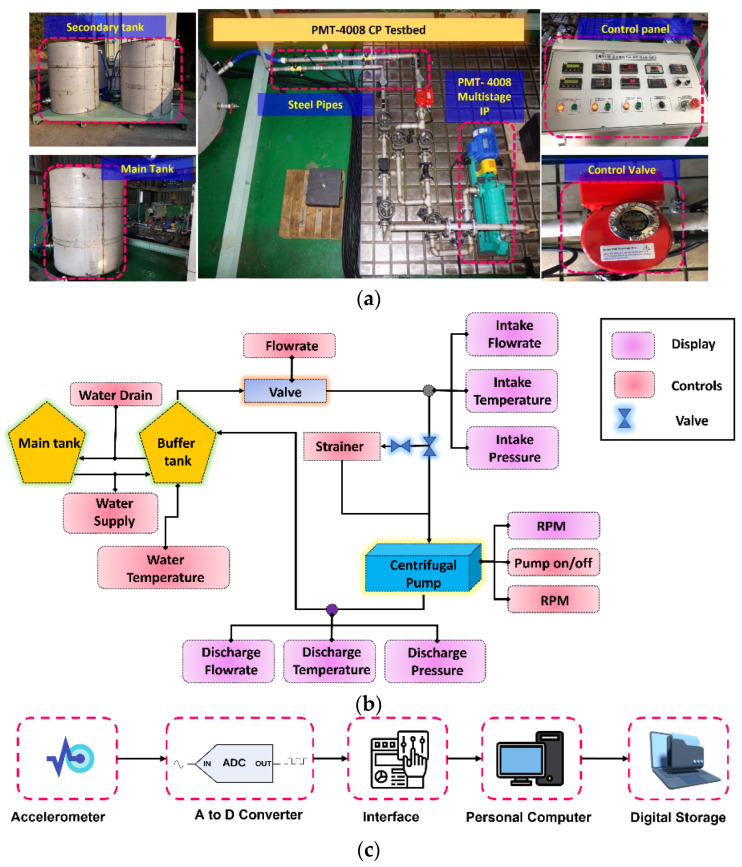
Data acquisition: (**a**) picture, (**b**) schematics, (**c**) system for data acquisition, and (**d**) SFs created in the CP.

**Figure 3 sensors-23-09090-f003:**
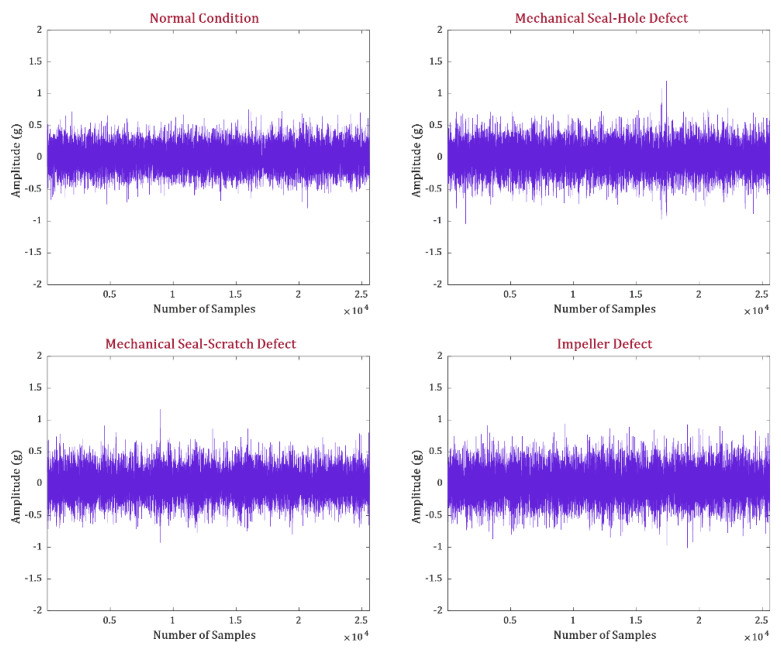
VS acquired under normal and faulty health states.

**Figure 4 sensors-23-09090-f004:**
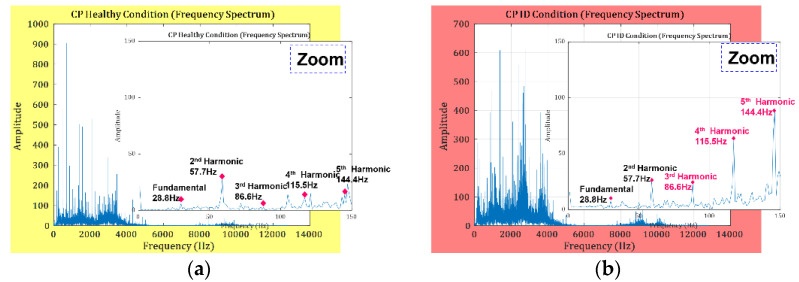
Frequency spectrum: (**a**) CP in healthy condition and (**b**) ID.

**Figure 5 sensors-23-09090-f005:**
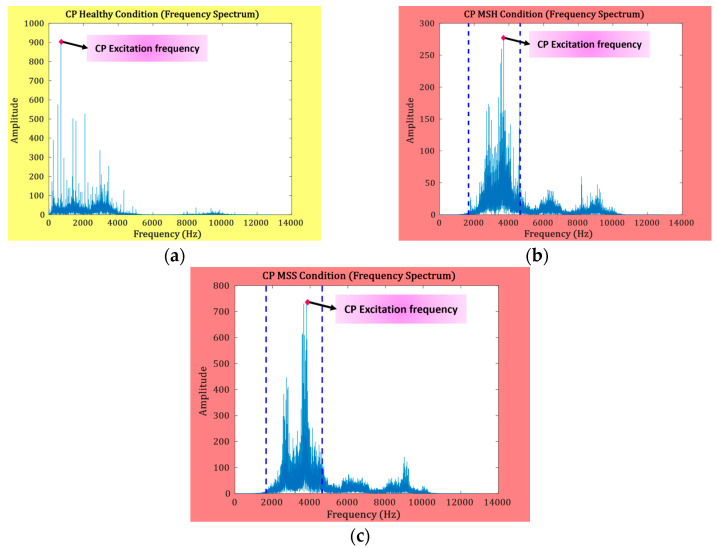
Frequency spectrum: (**a**) CP in healthy condition, (**b**) MS-H, and (**c**) MS-S.

**Figure 6 sensors-23-09090-f006:**
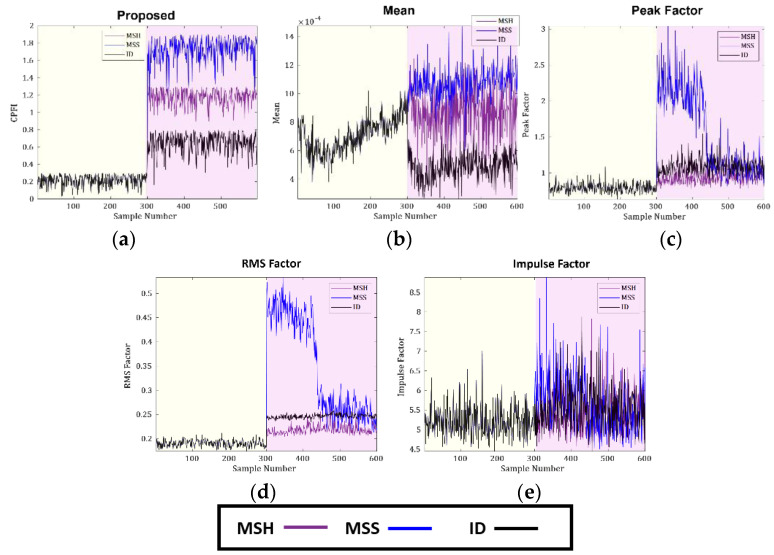
Comparison of the proposed CPFI with TD features for D-1: (**a**) proposed, (**b**) mean, (**c**) peak factor, (**d**) RMS factor, and (**e**) impulse factor.

**Figure 7 sensors-23-09090-f007:**
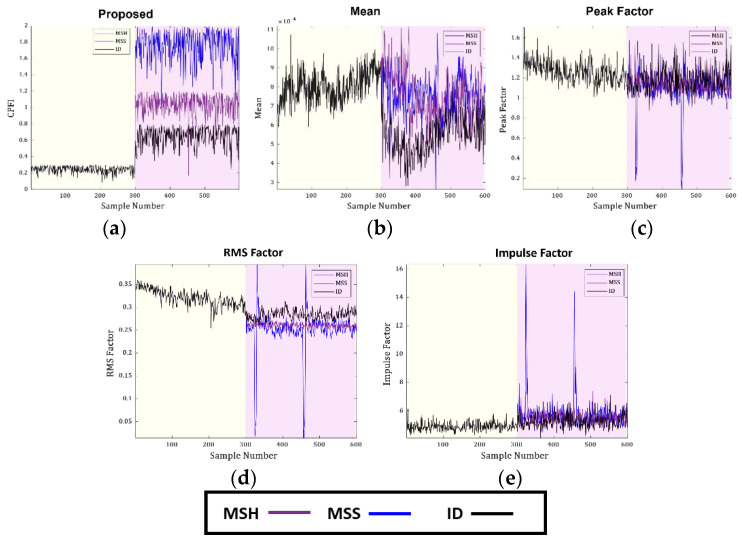
Comparison of the proposed CPFI with TD features for D-2: (**a**) proposed, (**b**) mean, (**c**) peak factor, (**d**) RMS factor, and (**e**) impulse factor.

**Figure 8 sensors-23-09090-f008:**
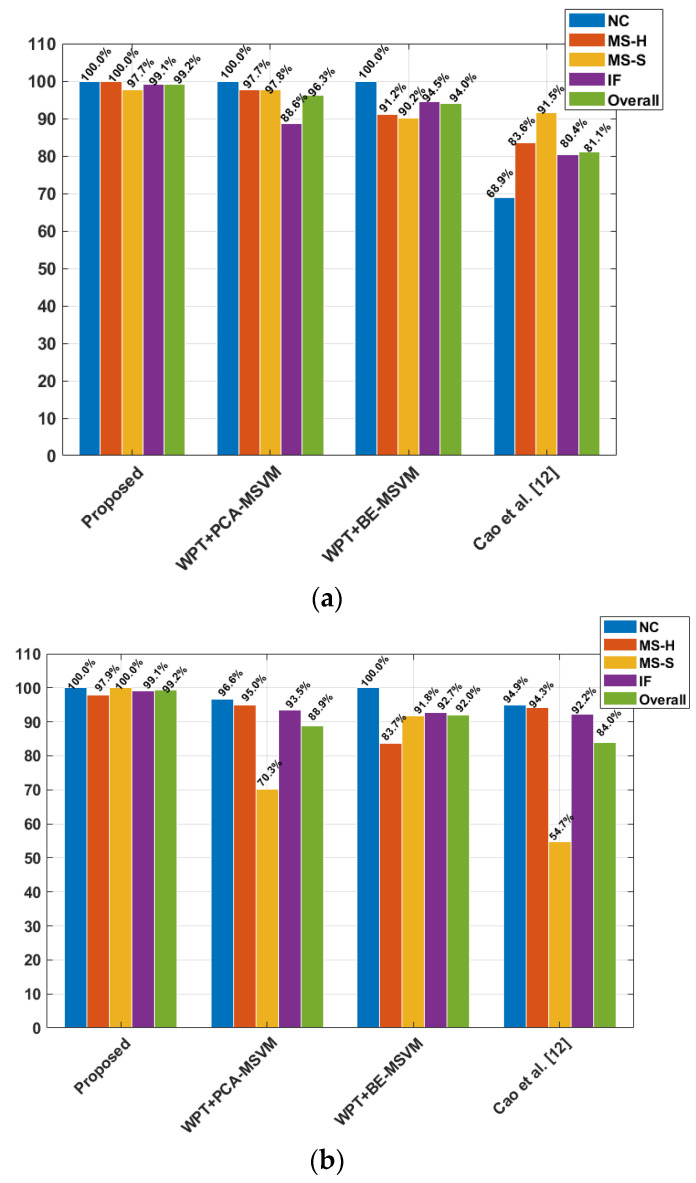
Per class TPR for proposed and reference methods (**a**) D-1 and (**b**) D-2.

**Table 1 sensors-23-09090-t001:** Description of D1 and D2.

Dataset	Normal/Defect	Pressure (bar)	Acquisition Time (min)	Total Samples
D1	Normal	3	5, 5, 5	300 (normal), 300, 300, 300
	MS-H			
	MS-S			
	ID			
D2	Normal	4	5, 5, 5	300 (normal), 300, 300, 300
	MS-H			
	MS-S			
	ID			

**Table 2 sensors-23-09090-t002:** Comparison between the proposed and reference techniques.

Dataset	Method	Precision	Recall	Error Rate
D-1	Proposed	99.20	99.21	1.56
	WPT-PCA-SVM	95.6	96.3	7.5
	WPT-BE-SVM	93.8	94.0	10.5
	Cao et al. [12]	79.8	81.1	25.4
D-2	Proposed	99.29	99.24	1.43
	WPT-PCA-SVM	89.75	88.87	17.13
	WPT-BE-SVM	92.09	92.04	13.29
	Cao et al. [12]	86.19	84.01	22.56

## Data Availability

The data were obtained from the industry. Owing to the privacy policy of the industry, the data are not publicly available.

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
