# Peer review of "A Technique for Centrifugal Pump Fault Detection and Identification Based on a Novel Fault-Specific Mann–Whitney Test"

_sensors, 2023, doi:10.3390/s23229090_

Round 1

Reviewer 1 Report

Comments and Suggestions for Authors

The authors must address all concerns to make sure that all issues affecting the sobriety of the research are improved/removed.

1-      English Writing: This article requires extensive proofreading, there are a lot of issues related to typos, grammar and spelling mistakes.

2-      All abbreviations should be defined when they first appear in the text.

3-      Abstract: It is very long and the word count should be reduced to less than 250 words. We recommend the authors to divide the abstract into purpose, methodology and numerical findings implicitly so that the reader can understand the research abstract comprehensively.

4-      The number of reviewed articles in the related works section is low, and the articles related to 2023 have not been reviewed.

5-      In the Related Work section should focus more on differences between this paper and other works to highlight the novelty of this paper. Also, the disadvantages and shortcomings of the previous methods that are addressed in the proposed method must be stated.

6-      The parameter of each equation must be described after using it. The parameters of some equations are not described.

7-      The quality of figures is poor; the author(s) must redraw them with high quality. Some text on figures is difficult to read.

8-      The descriptions given in this proposed scheme are not sufficient. Therefore, this paper needs more work. The proposed method should be described in detail, step by step. The key equations/models should be embedded in the approach.

9-      The complexity issue should be mentioned. Make a discussion about implementation requirements, giving numerical values such as memory capacity/CPU speed requirements.

10-   In conclusion the findings are not mentioned, give a brief definition of your findings.

Comments on the Quality of English Language

English Writing: This article requires extensive proofreading, there are a lot of issues related to typos, grammar and spelling mistakes.

Reviewer 2 Report

Comments and Suggestions for Authors

The paper presents a novel soft-fault detection and classification technique for centrifugal pumps (CPs) using a new fault-specific Mann-Whitney U-test and K-nearest neighbor (KNN) classification. It addresses the limitations of traditional features by developing a knowledge-independent Centrifugal Pump Fault Indicator (CPFI) based on the decomposition of vibration signatures into time-frequency representation using Wavelet Packet Transform (WPT). The proposed method demonstrated superior sensitivity to soft faults in CPs compared to traditional Time Domain (TD) indicators and other state-of-the-art methods, as validated through real-world experimental setups across varied operating conditions. The findings, illustrated through extensive comparative analysis and visualizations, indicate that the new method notably enhances soft-fault classification accuracy, showing promising potential for industrial applications, especially given its simplicity and low computational complexity. The paper also outlines the future direction of extending this method to diagnose other types of faults like fluid flow-related issues in CPs.

Major comments:

1. The paper does not provide a substantial discussion regarding the limitations of the proposed method.

2. The paper could benefit from a more extensive review of existing literature to situate its contributions within the broader scholarly and practical context.

3. The validation of the proposed method is based on specific experimental setups. There might be concerns regarding the generalizability of the findings to different or more challenging conditions, which were not addressed in the paper.

4. The comparative evaluation is a strong point of the paper, but the selection criteria for the reference methods used for comparison could be more transparent to ensure that there isn't any bias in portraying the proposed method favorably.

5. The paper could have elaborated more on the practical implications, ease of implementation, and the cost-effectiveness of the proposed technique in real-world scenarios.

6. The proposed method was validated using a specific dataset generated under controlled conditions. An external validation with datasets from different sources or real-world scenarios would have strengthened the claims of the paper.

7. The paper does not provide an analysis of how robust the proposed method is to various forms of noise, outliers, or non-standard conditions which are common in real-world operational environments.

8. While the paper provides statistical results, more rigorous statistical analysis could have been provided to substantiate the claims made. For instance, confidence intervals or p-values could be included to provide a sense of the statistical significance of the findings.

Comments on the Quality of English Language

There are some areas where the language could be refined for better clarity and readability. For instance, there might be instances of lengthy sentences that could be broken down for better understanding. Some technical terms or phrases could also be better explained or defined for readers who may not be familiar with the subject matter. Additionally, ensuring consistency in terminology and avoiding jargon or abbreviations without prior explanation could also improve the language quality.

Round 2

Reviewer 1 Report

Comments and Suggestions for Authors

After reviewing the amendments made by the authors that dealt with most of the reviewers' comments, the research appears better than the previous one. Accordingly, there is no objection to accepting the research, for the possibility of publishing it in the journal without additional modifications.

Author Response

Thanks for the positive response.

Reviewer 2 Report

Comments and Suggestions for Authors

The authors have addressed my comments.

Author Response

Thanks for the positive response.